# Searching for the Relationship between the Concentration of Heavy Metals in the Blood and the Clinical Course of Multiple Sclerosis: A Cross-Sectional Study in Poland

**DOI:** 10.3390/ijerph19116548

**Published:** 2022-05-27

**Authors:** Anna Knyszyńska, Karolina Skonieczna-Żydecka, Dorota Koziarska, Laura Stachowska, Artur Kotwas, Monika Kulaszyńska, Anna Lubkowska, Beata Karakiewicz

**Affiliations:** 1Department of Functional Diagnostics and Physical Medicine, Pomeranian Medical University in Szczecin, 71-210 Szczecin, Poland; anna.knyszynska@pum.edu.pl (A.K.); anna.lubkowska@pum.edu.pl (A.L.); 2Department of Biochemical Sciences, Pomeranian Medical University in Szczecin, Broniewskiego 24, 71-460 Szczecin, Poland; laura.stachowska@pum.edu.pl (L.S.); monika.kulaszynska@pum.edu.pl (M.K.); 3Department of Neurology, Pomeranian Medical University in Szczecin, Unii Lubelskiej 1, 72-252 Szczecin, Poland; dorota.koziarska@pum.edu.pl; 4Subdepartment of Social Medicine and Public Health, Department of Social Medicine, Pomeranian Medical University in Szczecin, Żołnierska 48, 71-210 Szczecin, Poland; artur.kotwas@pum.edu.pl (A.K.); beata.karakiewicz@pum.edu.pl (B.K.)

**Keywords:** heavy metals, cadmium, lead, environmental factors

## Abstract

(1) Background: Of all environmental pollutants, heavy metals have the most detrimental effect on public health because they remain in the ecosystem and are not biodegradable. The neurotoxicity of heavy metals, including cadmium and lead, has been well documented, and blood levels of heavy metals serve as a biomarker of exposure, reflecting their content in soft tissues. Multiple sclerosis (MS) is one of the most common chronic neurodegenerative diseases. The pathogenesis of MS is complex and relies on the interaction between genetic and environmental factors. The aim of this study was to search for the relationship between the values of cadmium and lead concentration in the blood and the health history and functional status of patients with MS. (2) Methods: The study group consisted of 151 patients with a clinical diagnosis of MS. Determination of the presence of tested elements in serum was performed using an ELAN DRC-e ICP mass spectrometer. (3) Results: Statistical analysis demonstrated that an increase in the level of cadmium was accompanied by an increase in the level of lead. There was no significant correlation between the blood lead concentration and the functional status assessed by EDSS among patients with MS in Poland. However, a tendency towards worse functional status of patients with higher blood lead concentration (*p* = 0.07) was demonstrated. Patients who experienced first MS symptoms at older age had higher blood levels of the tested metals. (4) Conclusions: The concentrations of cadmium and lead in the blood of MS patients in Poland were not factors differentiating their functional status and the course of the disease.

## 1. Introduction

Metal ions are required for many critical functions of the human body. Both deficiencies and excessive amounts of metal ions, even those necessary for normal human functioning, may lead to disease or poisoning of the body [1]. Of all environmental pollutants, heavy metals have the most detrimental effect on public health because they remain in the ecosystem and are not biodegradable. Exposure to heavy metals occurs from a variety of natural sources, including air, drinking water, food, and smoking [2]. Among others, lead (^208^Pb) and cadmium (^114^Cd) are metals that pose a considerable threat to human health. Widespread in the environment as a result of ecological imbalance, heavy metals are extremely harmful due to their ability to accumulate in living organisms and their chronic toxicity. Heavy metals also do not play any biological role in the human body. All their perceived effects are toxic, and they pose a serious threat to human health [3,4,5,6,7]. The molecular mechanism of lead toxicity is multifactorial, as it generates free radicals, reduces the sulfhydryl pools of glutathione antioxidants, inhibits enzyme activity, and blocks the uptake of important trace elements [8]. Thus, lead is believed to disrupt functions essential for neuronal homeostasis, including the inhibition of glycolytic enzymes in neurotransmitter metabolism [9]. Experimental evidence suggests that lead may cause white matter damage, changes in cellular structure, and even cell death [10]. Cadmium is the seventh most toxic heavy metal as ranked by the Agency for Toxic Substances and Disease Registry (ATSDR). The exact molecular mechanisms of cadmium toxicity remains unknown; however, it has been suggested that cadmium indirectly enhances the generation of free radicals and that it participates in oxidative stress through Fenton reaction [11]. Lead and cadmium toxicity has been observed to contribute to many important conditions, such as cancer, cognitive disorders, hypertension, heart diseases, and diabetes, as well as neurological disorders, including but not limited to multiple sclerosis (MS) [12]. 

One of the most common chronic neurodegenerative diseases is MS. It affects more than 2.5 million people worldwide. It is mainly diagnosed in adults between 20 and 45 years of age, more often in women and in the population of northern Europe [13,14]. The autoimmune background of this inflammatory disease allows for the identification of several main pathogenic factors, such as blood–brain barrier damage, the formation of multiple foci of perivascular cell infiltrates, myelin damage, loss of oligodendrocytes and axons, and secondary astroglial hyperplasia. However, the pathogenesis of MS is complex and relies on the interaction of genetic and environmental factors [15].

In the last decade, there has been a growing interest in the metabolism of neurotoxic metals and their impact on various neurodegenerative diseases; however, to date, their role has not been fully elucidated. Therefore, the aim of this study was to determine the relationship between the values of ^114^Cd and ^208^Pb concentration in the blood and the health history and functional status of patients with MS.

## 2. Materials and Methods

The study group consisted of 151 patients (47 men and 104 women) with a clinical diagnosis of MS under the care of the Provincial Center for Demyelinating Diseases in Szczecin. The patients’ disability was assessed by two neurologists at diagnosis in 2019 using the Kurtzke Expanded Disability Status Scale (EDSS). The scale was used to determine the overall mobility and functionality of the patients. It includes assessment of the major systems, i.e., the visual system, brainstem functions, motor and sensory impairment, cognitive function, and sphincter function. 

Additionally, a health interview was conducted regarding the EDSS score at the time of diagnosis, the age of the onset of symptoms, autoimmune diseases in the patient and their family history, the incidence of MS among the patient’s relatives, de novo MS, the presence of the first episode of MS, the number of affected systems, the form of the disease, and the number of foci at disease onset (monofocal or multifocal). 

The study was approved by the Bioethics Committee of Pomeranian Medical University in Szczecin (KB-0012/163/12).

### 2.1. Measurement Methodology

Determination of ^114^Cd and ^208^Pb in serum was performed using an ELAN DRC-e ICP mass spectrometer (PerkinElmer, Concord, ON, Canada). Before each assay, the instrument was tuned to achieve the optimal criteria recommended by the manufacturers. Oxygen was used as a reaction gas. The spectrometer was calibrated using an external calibration technique. Calibration standards were prepared fresh daily from 10 g/mL of multi-element calibration standard 3 (PerkinElmer Pure Plus, Shelton, CT, USA) by dilution with a blank reagent to final concentrations of 0.48, 0.99, and 1.98 g/L for ^114^Cd determination and 1, 2, 5, and 10 g/L for ^208^Pb determination. Correlation coefficients for calibration curves were always greater than 0.999. Matrix-matched calibration was applied. Rhodium was set as the internal standard.

The analysis protocol assumed a 30-fold dilution of serum in a blank reagent. The blank reagent consisted of high-purity water (>18 MΩ-cm), 0.5% TMAH (AlfaAesar, Kandel, Germany), 0.05% Triton X-100 (PerkinElemer, Shelton, CT, USA), 0.05% *n*-butanol (Merck, Darmstadt, Germany), 5 ug/L rhodium (PerkinElmer, Shelton, CT, USA), 200 ug/L gold (VWR, Steinheim, Germany), and 0.05% EDTA (Sigma-Aldrich, Leuven, Belgium). The accuracy and precision of all measurements were tested using a certified reference material (CRM), Clincheck Plasmonorm Blood Trace Elements Level 1 (Recipe, Munich, Germany). 

Recovery rates were between 80 and 105% for the analyzed elements; the calculated recurrency (Cv%) was below 15% for all of measured elements. The testing laboratory is a member of two independent external quality assessment schemes: LAMP, organized by CDC (LAMP: Lead and Multielement Proficiency Program; CDC: Center for Disease Control); and QMEQAS, organized by the Institut National de Santé Publique du Québec (QMEQAS).

### 2.2. Statistical Analysis

Statistical analysis was performed using MedCalc ver. 19.2 (Ostend, Belgium). The distribution of continuous variables was different from normal; therefore, the results were presented as median and quartile ranges. The differences in the tested continuous variables were analyzed using the Mann–Whitney U test or the Kruskal–Wallis test, and the Chi^2^ test was used for qualitative data. The significance level was set as *p* < 0.05, and *p* = 0.05–0.1 indicated a tendency toward statistical significance.

## 3. Results

### 3.1. Characteristics of the Study Sample 

The study group consisted of 151 subjects, including 104 women (68.9%) and 47 men (31.1%). Concomitant autoimmune diseases were only observed in 8.2% (*n* = 12) of the study sample, whereas 27.7% (*n* = 41) had a family history of such diseases. A family history of MS was found in 30.6% (*n* = 45) of patients, of whom 5.3% (*n* = 8) had a diagnosis of MS among collateral and 2.0% (*n* = 3) among direct relatives. De novo disease was found in 60.9% (*n* = 92) of the patients. Additionally, 41.1% (*n* = 62) of the subjects were characterized by a monofocal and 58.9% (*n* = 89) by a multifocal onset of the disease. In the course of the disease, 41.1% (*n* = 62) of the participants had one system affected, 41.7% (*n* = 63) had two systems, and the least—only 17.2% (*n* = 26)—had as many as three systems affected. 68.2% of the participants had been treated for more then 3 years. A total of 31.8% (*n* = 48) of the study participants had been treated for less than three years, but it should be noted that we were missing data on 4% (*n* = 6) of the patients. Primary attack of MS was observed in 36.4% (*n* = 55) of the patients. MS occurred in the study group in the following forms:96.70% (*n* = 146): relapsing–remitting multiple sclerosis (RRMS);2.60% (*n* = 4): secondary progressive multiple sclerosis (SPMS);0.70% (*n* = 1): primary progressive multiple sclerosis (PPMS).

The above clinical and sociodemographic parameters were analyzed in terms of sex. We observed that women diagnosed with MS achieved a significantly higher level of education than men with MS (number of education years: Me = 15.00 vs. Me = 13.00; *p* < 0.0001). Moreover, women were more likely to have attained an academic education (82.9% vs. 17.1%; *p* < 0.0001). The results are presented in Table 1 and Table 2.

### 3.2. Levels of the Tested Elements 

The reference values of the concentration of the analyzed metals in the blood were taken as the concentration of cadmium (5 µg/L) [16] and lead (31 µg/L) [17]. When analyzing the levels of ^114^Cd and ^208^Pb, we did not find differences between the sexes, although there was a statistical tendency for higher ^208^Pb levels in men (*p* = 0.0895). The results are shown in Figure 1 and Figure 2.

The relationships between the levels of lead and cadmium and EDSS scores at the time of diagnosis with disease, as well as EDSS score in 2019, the age at symptom onset, and the number of years of education, were examined. There were several correlations, which are presented in Table 3. Statistical analysis demonstrated that an increase in the level of cadmium was accompanied by an increase in the level of lead (r = 0.339, *p* < 0.0001). Moreover, a statistical tendency was observed for higher lead levels in subjects with higher EDSS scores at the time of diagnosis (r = 0.146, *p* = 0.0738) (Figure 3). We found that the higher the age at disease onset, the higher the lead and cadmium levels in patients (for cadmium, r = 0.186, *p* = 0.0224; for lead, r = 0.185, *p* = 0.0232). In turn, fewer years of education was correlated with higher cadmium levels (r = −0.161, *p* = 0.0480).

Subsequently, the association of cadmium and lead concentrations with clinical parameters and sociodemographic data was analyzed. The results are shown in Table 4 and Table 5. Statistical analysis revealed significantly higher lead levels in patients with a direct family history compared to those with de novo disease (*p* = 0.0453) (Figure 4). Moreover, there was a tendency for lead levels to increase with an increasing number of medications taken (*p* = 0.0870).

It was also found that people living in villages and cities with 25,000–100,000 inhabitants tended to have higher cadmium levels compared to their counterparts living in larger cities (Table 5). 

## 4. Discussion

In this paper, we present the clinical characteristics of 151 patients with multiple sclerosis in relation to the levels of lead and cadmium in their blood. These levels were 11.63 (4.34–49.49) μg/L and 0.35 (0.06–9.72) μg/L for the mean of lead and cadmium, respectively. Despite the lack of a control group of healthy people in the study, the results obtained in patients with MS in our Polish sample did not exceed the acceptable concentration levels of the analyzed elements in human biological material. Thus, they fell within the accepted ranges of the norm [16,17]. There were no gender differences in the levels of the tested heavy metals, although there was a trend towards higher lead levels in men. Earlier reports on lead levels in groups of MS patients were presented, among others, by Dehghanifiroozabadia et al. in 2019 [14] and Razavi et al. in 2016 [18]. However, there were large discrepancies between the results described by these authors. This may be due to the use of different methods for determining blood levels of metals or the geographic location of the area from which participants were recruited for the studies

To the best of our knowledge, there are only a few reports on the relationship between the levels of heavy metals (including lead and cadmium) and the clinical characteristics of MS patients. Such an evaluation was undertaken in 2016 by Razavi et al. [18], who analyzed patients living in Isfahan, a city in Iran. The results of our research show that individuals who experienced the first symptoms of the disease at an older age had higher levels of cadmium (r = 0.186; *p* = 0.024) and lead (r = 0.185; *p* = 0.023). Higher levels of one of the tested metals were accompanied by higher levels of the other element (r = 0.339; *p* < 0.0001). Analysis of the correlation between the levels of lead and the functional status of patients with MS indicates a tendency toworse functional status(higher EDSS scores) at the time of diagnosis with the disease in patients with higher levels of the tested elements in their blood. This relationship was not statistically significant (*p* = 0.07), but it was confirmed by other authors [18]. The discussed situation may result from the fact that lead has the ability to penetrate endothelial cells through the blood–brain barrier and that through a number of actions, it may affect both the central and peripheral nervous systems (especially motor nerves), leading to both structural and functional disorders [7]. Confirmation of the relationship between functional status and the levels of lead in the body of people with neurodegenerative diseases could be of considerable importance for both preventive measures and the treatment of patients.

The present study also demonstrated that patients with a family history of MS had significantly higher levels of lead than patients with de novo MS (*p* = 0.045). Although difficult to explain, we can assume that with increased exposure to lead in pregnant women (current or future MS patients), their babies could be more prone to MS due to longer exposure to environmental toxins, including lead [19]. However, no such correlation was observed in relation to cadmium levels (*p* = 0.510). Additionally, women reported a family history of MS more often than men (Chi^2^ = 5.210; *p* = 0.023) and were also more likely to have other immune diseases (Chi^2^ = 0.591; *p* = 0.015). Such a linkage between sex and MS has already been demonstrated [20], as well as the significantly higher incidence of MS [21] and other autoimmune diseases in women [22].

No significant relationships were observed between cadmium and lead levels and other assessed variables, such as the form of MS (*p* = 0.900) or place of residence (*p* = 0.09). Similar results were reported by Razavi et al. [18]. 

Previous findings and recent experimental studies have shown that exposure to heavy metals can enhance the progression of MS [18,23,24,25,26,27], and exposure to lead, owing to its ability to attach to myelin proteins and act as a hapten, is responsible for the formation of antibodies against myelin proteins, making it one of the environmental factors considered to play a significant role in the etiopathogenesis of MS [14]. However, relatively few epidemiological studies have been conducted to assess the correlation between MS and the levels of heavy metals, and the results of such studies are often inconsistent. Some authors have observed higher levels of lead and/or cadmium in the blood of MS patients compared to healthy individuals [14,28,29], whereas others did not observe such relationships [22,30,31,32,33]. Apart from studies confirming a higher incidence of MS in areas where soils contain high concentrations of lead [7,34,35,36], there are also studies showing no such relationships [37,38]. The only meta-analysis conducted to date in the field of toxic heavy metal concentrations in MS patients did not show any significant differences in blood lead levels between MS patients and healthy controls [39]. Therefore, the hypothesis about higher exposure to heavy metals among MS patients living in large cities in Poland has not been confirmed. However, it was not taken into account whether the city of residence was industrial or not.

The largest limitation of the study is the lack of a control group of healthy subjects; however, our main goal was to search for a relationship between the levels of heavy metals (lead and cadmium) and the variables characterizing MS patients. Moreover, it would be worth tracking changes in the levels of lead and cadmium in the course of MS, especially when observing changes in the functional status of patients. 

## 5. Conclusions

Despite the our earlier reports, the levels of cadmium and lead concentration in the blood were not determinants of the functional status of MS patients in Poland. However, this conclusion should be confirmed by studies on a larger sample including a control group of healthy people. Knowledge of the pathogenesis of neurodegenerative disorders, such as MS, is still incomplete, and their etiology remains unknown. However, proven neurotoxicity of heavy metals, including cadmium and lead, is a fact, and serum levels of heavy metals serve as a biomarker of exposure, reflecting their content in soft tissues. Hence, there is a need for further research in the field of blood lead levels in MS patients, taking into account a wider range of factors, extended mainly by the indicators of the functional status of patients.

## Figures and Tables

**Figure 1 ijerph-19-06548-f001:**
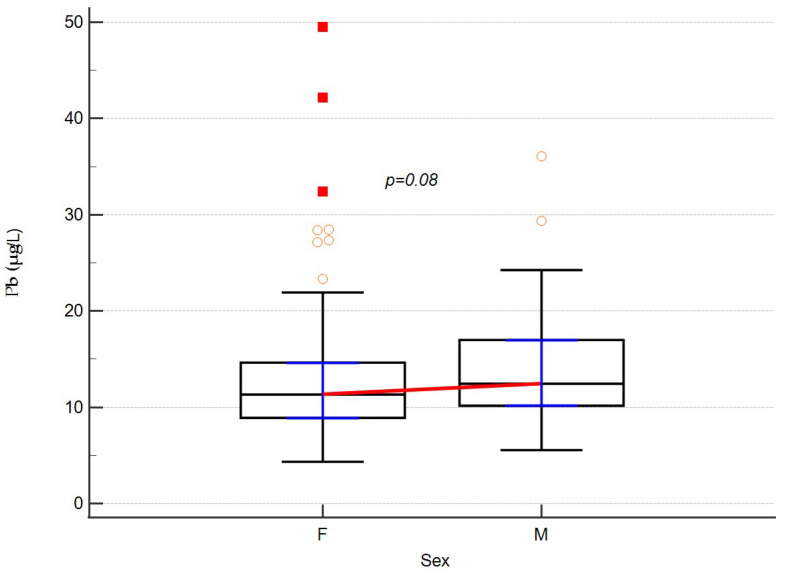
Lead levels with regard to sex. Medians and IQRs are shown. Empty dots represent individual data. Red squares are outliers.

**Figure 2 ijerph-19-06548-f002:**
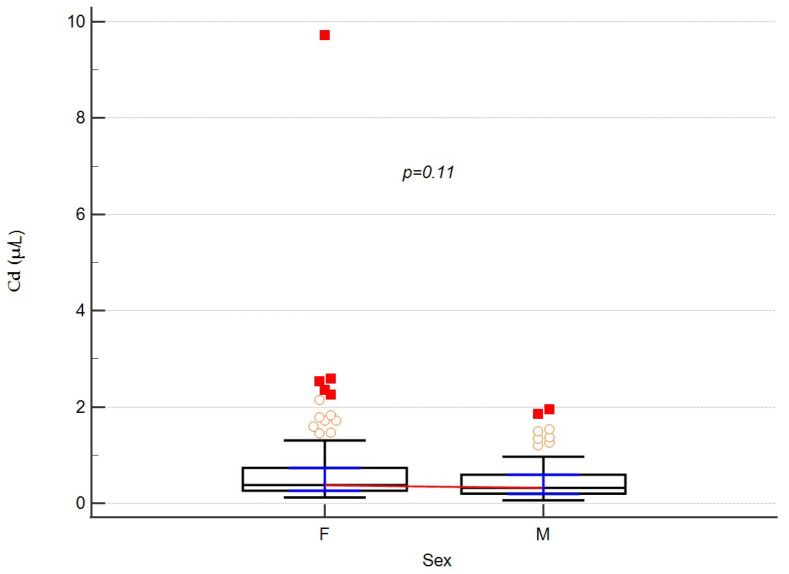
Cadmium levels with regard to sex. Medians and IQRs are shown. Empty dots represent individual data. Red squares are outliers.

**Figure 3 ijerph-19-06548-f003:**
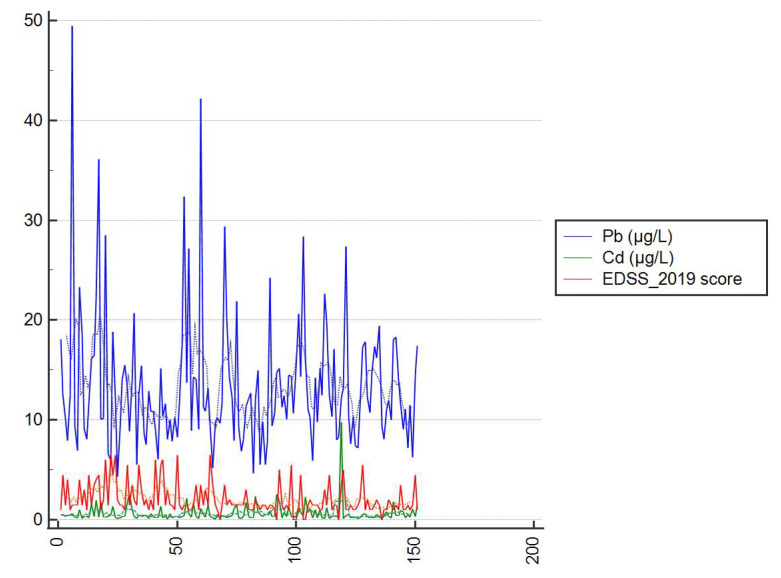
Correlation between the EDSS-BL scores and the levels of the tested elements. Dotted lines represent trendlines.

**Figure 4 ijerph-19-06548-f004:**
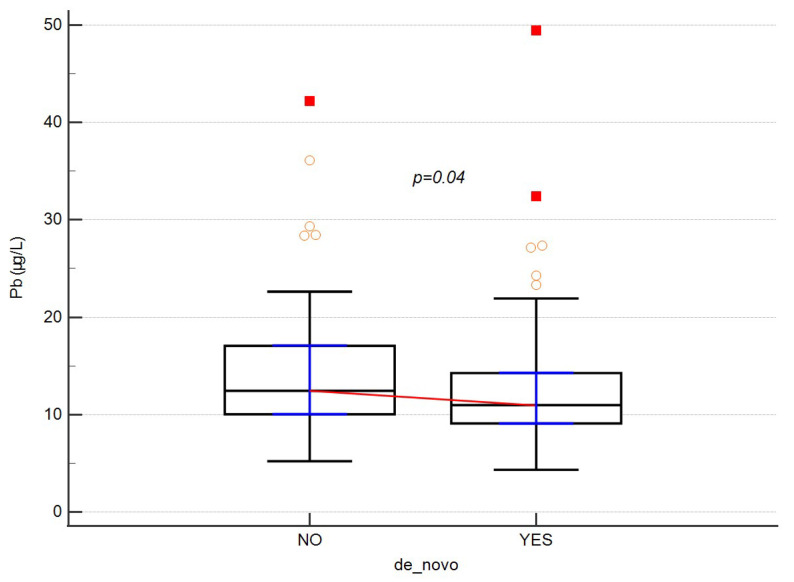
Levels of lead with regard to disease onset. Empty dots represent individual data. Red squares are outliers.

**Table 1 ijerph-19-06548-t001:** Clinical and sociodemographic data of MS patients with regard to sex.

Clinical Parameters and Sociodemographic Data	Sex						
	Women			Men			*p*
	*n*	Me	M	*n*	Me	M	
EDSS 2019 (scores)	104	1.5	1.99	47	1.5	2.30	0.58
EDSS result at time of diagnosis (scores)	104	1.5	1.72	47	1.5	1.95	0.28
Age at symptom onset (years)	104	29.5	31.11	47	27	29.28	0.15
Duration of education (years)	104	15	15.00	47	13	13.19	0.0002

*n*—number, M—mean, Me—median, *p*—statistical significance.

**Table 2 ijerph-19-06548-t002:** Qualitative clinical and sociodemographic data with regard to sex.

Clinical Parameter		Sex		Chi^2^	*p*
		Women	Men		
		*n*	*n*		
Autoimmune diseases	No	89	46	ne	ne
	Yes	12	0		
De novo	No	38	21	0.90	0.34
	Yes	66	26		
Primary MS attack	No	66	30	0.002	0.96
	Yes	38	17		
Number of systems affected	One	42	20	2.18	0.34
	Two	41	22		
	Three	21	5		
Disease onset	Monofocal	42	20	0.06	0.80
	Multifocal	62	27		
Form of disease	PPMS	0	1	ne	ne
	RRMS	101	45		
	SPMS	3	1		
Education	Primary	1	3	25.17	<0.0001
	Vocational	3	10		
	Secondary	32	20		
	Higher	68	14		
Place of residence	Village	15	4	2.52	0.47
	City of 25,000–100,000 inhabitants	10	7		
	City > 100,000 inhabitants	14	9		
	City ≤ 25,000 inhabitants	65	27		

*n*—number of subjects in the group, (%)—percentage, *p*—statistical significance, ne—not estimable.

**Table 3 ijerph-19-06548-t003:** Correlations between the levels of the tested elements and selected clinical and demographic variables. ne—not estimated.

Clinical Parameter	Element			
	^114^Cd [µg/L]		^208^Pb [µg/L]	
	r	*p*	r	*p*
EDSS 2019	0.04	0.66	0.08	0.32
EDSS score at time of diagnosis	0.03	0.70	0.15	0.07
Age at symptom onset	0.19	0.02	0.18	0.02
Duration of education [years]	−0.16	0.05	−0.07	0.41
^114^Cd level [µg/L]	ne	ne	0.34	<0.0001
^208^Pb level [µg/L]	0.34	<0.0001	ne	ne

**Table 4 ijerph-19-06548-t004:** Relationships between clinical parameters and the levels of the tested elements.

Element													
Clinical Parameter		^114^Cd						^208^Pb					
		*n*	Min	Max	Me	25–75	*p*	*n*	Min	Max	Me	25–75	*p*
Autoimmune diseases	No	135	0.06	9.72	0.35	0.22–0.62	0.38	135	4.34	49.49	11.68	9.41–15.16	0.28
	Yes	12	0.14	2.59	0.42	0.33–0.85		12	5.92	28.38	10.30	7.91–13.63	
De novo	No	59	0.13	9.72	0.35	0.22–0.55	0.48	59	5.22	42.20	12.45	10.04–17.07	0.04
	Yes	92	0.06	2.59	0.36	0.25–0.36		92	4.34	49.49	10.97	9.09–14.27	
First MS attack	No	96	0.12	2.59	0.34	0.23–0.61	0.59	96	4.34	49.49	11.66	9.10–16.35	0.69
	Yes	55	0.06	9.72	0.38	0.31–0.92		55	5.52	42.20	11.62	9.57–14.29	
Number of systems affected	One	62	0.15	2.54	0.36	0.25–0.59	0.49	62	4.34	49.49	10.98	9.41–14.35	0.39
	Two	63	0.06	9.72	0.32	0.21–0.55		63	5.55	32.41	12.27	9.14–15.16	
	Three	26	0.14	2.59	0.38	0.26–1.08		26	5.93	36.10	12.94	9.12–16.45	
Disease onset	Monofocal	62	0.15	2.54	0.36	0.25–0.59	0.75	62	4.34	49.49	10.98	9.41–14.35	0.18
	Multifocal	89	0.06	9.72	0.34	0.23–0.84		89	5.55	36.10	12.45	9.12–15.64	
Form of disease	PP	1	0.44	0.44	0.44	0.44–0.44	0.90	1	12.45	12.45	12.45	12.45–12.45	0.89
	RR	146	0.06	9720	0.35	0.23–0.35		146	4.34	49.49	11.61	9.35–15.15	
	SP	4	0.22	1.31	0.300	0.25–0.30		4	8.29	28.48	13.92	8.64–23.67	
Number of medications taken	One	82	0.08	9.72	0.35	0.23–0.81	0.98	82	4.34	49.49	12.50	9.41–16.32	0.09
	Two	40	0.14	2.14	0.36	0.24–0.61		40	5.22	32.41	10.85	9.27–12.45	
	Three	9	0.13	1.86	0.43	0.17–0.54		9	5.93	22.62	15.16	11.68–16.65	
	Four	13	0.06	2.59	0.32	0.20–1.21		13	4.67	15.15	10.08	8.64–11.981	
	No data	6	0.22	1.31	0.32	0.27–0.44		6	8.29	28.48	11.14	8.99–18.85	

**Table 5 ijerph-19-06548-t005:** Relationships between sociodemographic data and the levels of the tested elements.

Element													
Clinical Parameters		^114^Cd						^208^Pb					
		*n*	Min	Max	Me	25–75	*p*	*n*	Min	Max	Me	25–75	*p*
Place of residence	Village	19	0.162	2.35	0.47	0.24–1.01	0.09	19	7.40	49.49	12.00	9.86–19.41	0.56
	City 25,000–100,000 inhabitants	17	0.182	1.34	0.41	0.24–0.75		17	4.67	36.10	10.77	7.98–12.98	
	City > 100,000 inhabitants	23	0.06	2.59	0.59	0.32–1.15		23	7.83	42.20	11.34	9.85–14.71	
	City ≤ 25,000 inhabitants	92	0.08	9.72	0.33	0.21–0.53		92	4.34	32.41	12.19	9.10–15.29	
Education	Primary	4	0.15	1.259	0.285	0.21–1.26	0.24	4	7.40	14.47	10.68	8.25–13.37	0.23
	Vocational	13	0.06	1.95	0.32	0220–1.95		13	9.80	28.48	15.15	10.20–20.44	
	Secondary	52	0.15	9.72	0.41	0.31–0.97		52	4.34	49.49	11.80	9.42–15.86	
	Higher	82	0.08	2.59	0.34	0.21–0.55		82	5.22	42.20	11.42	8.58–14.51	

## Data Availability

Data are available upon reasonable request.

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
