# Peer review of "Searching for the Relationship between the Concentration of Heavy Metals in the Blood and the Clinical Course of Multiple Sclerosis: A Cross-Sectional Study in Poland"

_ijerph, 2022, doi:10.3390/ijerph19116548_

Round 1
Reviewer 1 Report
The manuscript ‘Heavy Metals and Multiple Sclerosis – is there a Link?’ by KnyszyÅ„ska et al, describes a possible link between increased levels of heavy metals and the development of MS disease. Authors have examined Pb and Cd levels in MS patients using ICP mass spectrometry and concluded to act these levels as biomarkers for the disease. The manuscript is written well. The investigation strategy is well designed but missing data on healthy individuals.
The specific comments, which could help to improve the manuscript, are:
- Pd and Cd representation in the manuscript has to be corrected as 114Cd and 208Pb.
- There is no mention of the basal levels of heavy atoms in a healthy individual (particularly Pb and Cd).
- Ambiguities are there in representing decimal numbers throughout the manuscript. e.g. Line 96; 0,48;0,99 and Line 140; 96.7%.
- Line 101: purity of water unit is MΩ-cm, not MW. What is the stoichiometry of individual components in blank reagent?
- In the statistical section on page 3: Insignificant text is present Line119-Line124.
- Figure 3,4: No label on X and Y axes.
- Line 186,187: “people living in 1 and 2” What are these 1 and 2?.
- Line 194: “neither….of humans” sentence needs to be corrected.
- Chi2 should be represented as Chi2.
Author Response
We wish to thank both the Editor and our Reviewers for the thoughtful and detailed comments on our manuscript. The comments have improved our manuscript by far. We have responded to each concern below.
Reviewer 1:
The manuscript ‘Heavy Metals and Multiple Sclerosis – is there a Link?’ by KnyszyÅ„ska et al, describes a possible link between increased levels of heavy metals and the development of MS disease. Authors have examined Pb and Cd levels in MS patients using ICP mass spectrometry and concluded to act these levels as biomarkers for the disease. The manuscript is written well. The investigation strategy is well designed but missing data on healthy individuals.
The specific comments, which could help to improve the manuscript, are:
- Pd and Cd representation in the manuscript has to be corrected as 114Cd and 208Pb.
A: These were amended in the text.
- There is no mention of the basal levels of heavy atoms in a healthy individual (particularly Pb and Cd).
A: Such information was added
- Ambiguities are there in representing decimal numbers throughout the manuscript. e.g. Line 96; 0,48;0,99 and Line 140; 96.7%.
A: This was amended throughout the text
- Line 101: purity of water unit is MΩ-cm, not MW. What is the stoichiometry of individual components in blank reagent?
A: This was amended and new information added in the Methods section.
- In the statistical section on page 3: Insignificant text is present Line119-Line124.
A: I apologize for that, this was left from template while preparing submission. It was deleted.
- Figure 3,4: No label on X and Y axes.
A: The labels were added
- Line 186,187: “people living in 1 and 2” What are these 1 and 2?.
A: The proper text was included
- Line 194: “neither….of humans” sentence needs to be corrected.
A: The text was amended, as requested
- Chi2 should be represented as Chi2.
A: This was amended throughout the text
Reviewer 2 Report
In this manuscript, several drawbacks exist and many concerns need to be addressed as follows:
- The title seems like the title of a review not an original article. It is highly recommended to edit the title to be representative of the study and clarify the country of the study.
- More representative keywords should be added.
- Abstract:
- Line 20: replace 114Cd and 208Pb with Cd and Pb to avoid confusion.
- The conclusion is general information. Please rewrite the conclusion of the specific findings of the study.
- Introduction:
- It should end with the objective of the study.
- As the authors mentioned mercury is one of the main elements that could affect human health. Why the authors have not analyzed mercury in the serum samples?
- Material and methods:
- Collect the points “lines 67-84” into one paragraph.
- Add the reference for the method of heavy metal analysis.
- Lines 122-124: “Interventionary studies involving animals or humans, and other studies that require ethical approval, must list the authority that provided approval and the corresponding ethical approval code”. This is the instructions of the journal or what?
- Results:
- Figure1 1 and 2: reconstructive with another more scientific diagram. Also, the unit of measurement of the analyzed metal should be added to the Y-axis.
- The discussion needs to be extensively revised as, in the present form, it is just a repetition of the introduction and results. Several points are confusing as follows:
- The authors mentioned that “Neither the mean nor the median values did not exceed the levels acceptable in biological material of humans” ”lines 194-195”. This means that the biological values of cadmium and lead in human fluid need to be reevaluated or what?
- 210-213: the authors mentioned that “Analysis of the correlation between the levels of lead and the functional status of the patients with MS was not statistically significant”. This should be clearly defined in both the abstract and conclusion of the study.
- 221-223: what is the possible attribution for the positive correlation between the family history of MS and the higher levels of lead?
- Conclusion in the present form is a future perspective. Where is the conclusion of the findings of the study?
- It is not preferred to begin sentences with abbreviations like MS in lines 17 and 56. Please revise the whole manuscript for such an error.
- The manuscript needs to be revised for the English and the overall writing style. The writing style should be formal from the third-person perspective. Do not use we or our.
- There is a problem in using abbreviations throughout the manuscript. The full term should be mentioned first with the abbreviation between paresis then the abbreviations should be exclusively used throughout the manuscript. E.g., in line 17, MS should be replaced with Multiple sclerosis (MS) then the abbreviation should be exclusively used further. Such errors have been repeated for many abbreviations throughout the manuscript.
Author Response
We wish to thank both the Editor and our Reviewers for the thoughtful and detailed comments on our manuscript. The comments have improved our manuscript by far. We have responded to each concern below.
Reviewer 2
In this manuscript, several drawbacks exist and many concerns need to be addressed as follows:
- The title seems like the title of a review not an original article. It is highly recommended to edit the title to be representative of the study and clarify the country of the study.
A: The new title including study design and country was proposed
- More representative keywords should be added.
A: We included two more key words and left one
- Abstract:
- Line 20: replace 114Cd and 208Pb with Cd and Pb to avoid confusion.
A: This was amended as requested
- The conclusion is general information. Please rewrite the conclusion of the specific findings of the study.
A: We have corrected the conlusions as advised.
- Introduction:
- It should end with the objective of the study.
A: We added the aim as requested.
- As the authors mentioned mercury is one of the main elements that could affect human health. Why the authors have not analyzed mercury in the serum samples?
A: We decided to delete this information, as – in fact – there was no necessity to include this into the manuscript.
- Material and methods:
- Collect the points “lines 67-84” into one paragraph.
A: This was done ad advised.
- Add the reference for the method of heavy metal analysis.
A proper reference was added
- Lines 122-124: “Interventionary studies involving animals or humans, and other studies that require ethical approval, must list the authority that provided approval and the corresponding ethical approval code”. This is the instructions of the journal or what?
A: A: I apologize for that, this was left from template while preparing submission. It was deleted.
- Results:
- Figure1 1 and 2: reconstructive with another more scientific diagram. Also, the unit of measurement of the analyzed metal should be added to the Y-axis.
A: The figures were replaced with box-plots and units were added.
- The discussion needs to be extensively revised as, in the present form, it is just a repetition of the introduction and results. Several points are confusing as follows:
- The authors mentioned that “Neither the mean nor the median values did not exceed the levels acceptable in biological material of humans” ”lines 194-195”. This means that the biological values of cadmium and lead in human fluid need to be reevaluated or what?
A: The text was rephrased to avoid misunderstanding and illegibility
- 210-213: the authors mentioned that “Analysis of the correlation between the levels of lead and the functional status of the patients with MS was not statistically significant”. This should be clearly defined in both the abstract and conclusion of the study.
A: such text was added into the abstract section
- 221-223: what is the possible attribution for the positive correlation between the family history of MS and the higher levels of lead?
A: A possible explanation was added in the discussion
- Conclusion in the present form is a future perspective. Where is the conclusion of the findings of the study?
A: Conclusion was rewritten as advised
- It is not preferred to begin sentences with abbreviations like MS in lines 17 and 56. Please revise the whole manuscript for such an error.
A: We amended that. Thank you.
- The manuscript needs to be revised for the English and the overall writing style. The writing style should be formal from the third-person perspective. Do not use we or our.
A: The manuscript was evaluated by professional English editing bureau. The certificate was obtained.
- There is a problem in using abbreviations throughout the manuscript. The full term should be mentioned first with the abbreviation between paresis then the abbreviations should be exclusively used throughout the manuscript. E.g., in line 17, MS should be replaced with Multiple sclerosis (MS) then the abbreviation should be exclusively used further. Such errors have been repeated for many abbreviations throughout the manuscript.
A: I apologize for that. We made our best to change these mistakes.
Round 2
Reviewer 2 Report
No further comments to be addressed